# The Influence of Whole-Body Cryotherapy or Winter Swimming on the Activity of Antioxidant Enzymes

**DOI:** 10.3390/biology13050295

**Published:** 2024-04-25

**Authors:** Bartłomiej Ptaszek, Szymon Podsiadło, Olga Czerwińska-Ledwig, Aneta Teległów, Wanda Pilch, Ewa Sadowska-Krępa

**Affiliations:** 1Institute of Applied Sciences, University of Physical Education in Krakow, 31-571 Krakow, Poland; 2Institute of Clinical Rehabilitation, University of Physical Education in Krakow, 31-571 Krakow, Poland; szymon.podsiadlo@awf.krakow.pl; 3Institute of Basic Sciences, University of Physical Education in Krakow, 31-571 Krakow, Poland; olga.czerwinska@awf.krakow.pl (O.C.-L.); aneta.teleglow@awf.krakow.pl (A.T.); wanda.pilch@awf.krakow.pl (W.P.); 4Institute of Sport Sciences, The Jerzy Kukuczka Academy of Physical Education, 40-065 Katowice, Poland; e.sadowska-krepa@awf.katowice.pl

**Keywords:** whole-body cryotherapy, winter swimming, oxidative stress, CAT, GPx, SOD

## Abstract

**Simple Summary:**

Whole-body cryotherapy and winter swimming (cold bath) are treatments that use low temperature as a therapeutic stimulus and have become increasingly popular in recent years. These treatments have many positive effects on the human body. The most important reactions of the body include analgesic and anti-swelling effects, as well as reactions of the immune and circulatory systems and changes in the concentration of many hormones. However, some reports indicate that exposure to extremely low temperatures may induce the production of reactive oxygen species in the body. Healthy people often have an increased level of oxidative stress, which may cause abnormal endothelial function and, consequently, an increased risk of developing hypertension, atherosclerosis and other diseases. Oxidative stress can likewise cause acute or chronic inflammation. In order to combat the harmful effects of free oxygen radicals, in this study we assessed the level of antioxidant enzymes in patients after receiving a cold stimulus.

**Abstract:**

The aim of this study was to investigate the effect of a series of 20 whole-body cryotherapy sessions and 20 winter swimming sessions on the level of catalase, glutathione peroxidase and superoxide dismutase. The experimental group consisted of 60 people (30 people received cryotherapy and 30 people swam in cold water). The control group—without intervention: 30 people. Each of the three groups was tested twice. Analyzing the changes in the examined indicators, a statistical increase of CAT was observed in men after the use of WBC, as well as an increase of SOD in women and a decrease of SOD in men after the winter swimming season. Regular WS does not seem to place an excessive burden on the body in terms of intensifying oxidative processes. WS sessions once a week can be recommended as an effective method of improving health by causing positive adaptive changes and strengthening the body’s antioxidant barrier. WBC may be used as an adjunct therapy in the treatment of diseases caused by oxidative stress, as it improves the body’s antioxidant capacity. Further research is needed to elucidate antioxidant mechanisms in humans and to determine the short- and long-term effects of exposure to WS and WBC.

## 1. Introduction

Whole-body cryotherapy (WBC) and winter swimming (WS) are treatments that use low temperature as a therapeutic stimulus and have become more popular in recent years. The physiological reactions of the body occurring after a series of WBC treatments include, among others, analgesic and anti-swelling effects as well as reactions on the part of the immune and circulatory systems [1,2]. WBC also causes changes in the concentration of such hormones (corticosteroids, beta-endorphins, norepinephrine) [3]; however, some reports indicate that exposure to extremely low temperatures may induce the production of reactive oxygen species (ROS) in the body [4]. However, according to the hormesis theory, the antioxidant effect of WBC is also postulated [5] and neutralization of oxidative stress is considered to be a key mechanism that can explain the positive impact of cryotherapy [6]. Healthy people often have an increased level of oxidative stress, which may cause abnormal endothelial function and, consequently, an increased risk of developing hypertension, atherosclerosis and other diseases [7]. Oxidative stress can likewise cause acute or chronic inflammation [8]. In order to combat the harmful effects of free oxygen radicals, cells contain antioxidant enzymes, including catalase (CAT), superoxide dismutase (SOD) and glutathione peroxidase (GPx) [9,10].

WS, also known as cold water swimming, often leads to immediate as well as long-term physiological and biochemical reactions, including both hormonal and metabolic reactions, as well as reactions of the cardiovascular system [11,12,13]. Stimulation of the sympathetic nervous system and adrenal glands, the observed increase in the concentration of catecholamines and, to a lesser extent, adrenaline, stimulates thermogenesis, regulates vasoconstriction and, together with cortisol, takes part in energy metabolism [14,15]. Few studies have confirmed that swimming in winter causes oxidative stress, and repeated immersion in cold water may enhance the immune response and improve antioxidant protection [16,17,18,19]. It is also known from previous studies that acute exercise and cold exposure increase the level of peroxisome proliferator-activated receptor γ coactivator-1 (PGC-1α) in muscle and adipose tissue [20,21], which leads to an increase in antioxidant defense [22].

The aim of this study was to investigate and compare the effects of a series of 20 WBC sessions and 20 WS sessions on the levels of CAT, GPx and SOD. It was hypothesized that WBC and WS would improve the antioxidant capacity of the body in healthy people, measured by the following indicators: CAT, GPx, SOD. Pro-oxidative-antioxidant processes play an important role in the development of several different pathologies, which can also cause adaptive changes that protect tissues against pro-antioxidant imbalances. Understanding the relationship between vascular damage, neuroinflammation and oxidative stress is fundamental to understanding the pathogenesis of neurodegenerative diseases.

## 2. Materials and Methods

### 2.1. Participant Characteristics

The presented prospective, controlled study, is consistent with the assumptions of the Helsinki Declaration, with the approval number of the Bioethical Committee of the District Medical Chamber in Krakow: 194/KBL/OIL/2019 on 17 September 2019. Each volunteer read the information about the study design and was given the opportunity to ask questions, after which they gave informed written consent to participate in the study. A rehabilitation doctor and a physiotherapist took care of the participants’ safety.

Inclusion Criteria: age: 30–55 years; very good overall health (without chronic diseases); written consent to participate in this study. Exclusion Criteria: participation in other forms of physical activity directly before or during this study; changing the diet immediately before or during the project; contraindications to WBC or WS.

WBC group: 15 women and 15 men; WS group: 15 women and 15 men; control group—without intervention: 15 women and 15 men (Table 1).

### 2.2. Blood Sample Analysis

For the analysis of blood biochemical parameters, venous blood was collected twice:Study 1: on the day of the commencement of WBC or at the beginning of the WS season (November);Study 2: after a series of 20 cryotherapy sessions or at the end of the WS season (March).

The control group was also tested twice (four-week break).

Blood was collected in tubes with EDTA in the morning from the cephalic, fallen or medial vein. The blood was centrifuged at 1000× *g* for 10 min at 4 °C to separate plasma and erythrocytes, which were then washed three times with cold (4 °C) saline and kept frozen at −80 °C until analyzed for the activity of antioxidant enzymes: catalase (CAT, EC 1.11.1.6, Aebi’s method [23]); glutathione peroxidase (GPx, EC 1.11.1.9, the commercial RANSEL RS504 kit by Randox, Crumlin, UK) and superoxide dismutase (SOD, EC 1.15.1.1, the commercially available RANSOD SD125 kit by Randox, UK). Biochemical assays were performed by a laboratory certified as meeting the requirements of PN-EN ISO 9001:2015 [24], in line with the recommendations of the testing kits manufacturers. The same procedure was used in our previous studies [25].

### 2.3. Description of the Intervention

WBC treatments were performed at the Małopolska Cryotherapy Rehabilitation Center in Krakow. The temperature of the atrium in the cryogenic chamber during the procedure: −60 °C; the temperature of the cryogenic chamber during the procedure: −120 °C. The treatments were performed in a Wrocław-type cryochamber, where liquid nitrogen is used for cooling. The duration of a single cryotherapy session was 3 min and the stimulus was graded (1.5 min—1st treatment; 2 min—2nd treatment). One treatment was performed a day (every day at the same time between 3:00 p.m. and 5:00 p.m.). A total of 20 treatments took place, which were performed 5 times a week in the winter months.

A single WS session lasted 4 min (November–March)—1x/week—20 baths per season. Cold water swimming parameters: water temperature 3–6 °C. The baths were carried out at the Kraków Winter Swimming Club “Kaloryfer” (Radiators)—Bagry Lagoon. The bath consisted of whole-body immersion in a cold lake (excluding the head). During the bath, the subjects moved their upper and lower limbs while submerged.

Completely different but most commonly used WBC and CWS standard protocols are presented [25,26,27,28,29,30,31,32,33,34,35,36,37,38,39,40,41].

### 2.4. Statistical Analysis

Descriptive statistics were determined: mean (x) as well as standard deviation (SD). The normality of distributions was verified with the Shapiro–Wilk test. Data distribution analysis was performed using parametric tests—the Student’s *t*-test for dependent samples (Study 1 vs. Study 2). The applied tests verified two-sided hypotheses. Comparisons within groups depending on the intervention were compared by anova analysis of variance. When the result of the analysis of variance showed significant differences, post hoc tests were performed. The analyses were performed with the use of the Statistica 13 package (Tibco Software Inc., Palao Alto, CA, USA).

## 3. Results

Comparisons performed using the Student’s *t*-test for dependent groups showed several significant changes within the study groups depending on the intervention performed. After a series of WBC procedures, a statistically significant increase of the CAT level [U/g Hb] was observed in the group of men (Table 2). In the case of WS intervention, significant changes were observed in the SOD parameter [U/g Hb]. In the women’s group, the Student’s *t*-test showed a significant statistical increase in the examined indicator, and in the men’s group, a statistically significant decrease in the level of this parameter (Table 3). In the control groups, the statistical analysis performed did not show any changes in the examined parameters over time (Table 4). One-way analysis of variance showed statistically significant differences between the interventions performed (WBC vs. WS vs. CONT). In the group of women, the differences concerned the SOD [U/g Hb] index and in the men’s group CAT parameter [U/g Hb] (Table 5). Post hoc tests for the SOD [U/g Hb] parameter in the women’s group showed statistically significant differences between the WBC and WS groups and the WS and CONT groups (Table 6). However, in the group of men, post hoc tests for the CAT parameter [U/g Hb] showed statistically significant differences between the WBC and WS groups and the WBC and CONT groups (Table 7).

## 4. Discussion

This study attempted to assess the effects of a series of 20 WBC sessions and 20 WS sessions on antioxidant enzymes (CAT, GPx, SOD) in healthy women and men. Analyzing the changes in the examined indicators, a statistical increase of CAT was observed in men after the use of WBC, as well as an increase of SOD in women and a decrease of SOD in men after the winter swimming season. Oxidative stress is implicated with neurodegenerative diseases. This is due to a long-term change in metabolism, exposure to exogenous factors or oxidizing compounds, and is associated with an inflammatory reaction [26,30]. Pro-oxidative-antioxidant processes play an important role in the development of several different pathologies, which can also cause adaptive changes that protect tissues against pro-antioxidant imbalances [31].

The results of the research by Lubkowska et al. (2009) showed that one cryostimulation treatment causes oxidative stress in healthy people, although its level is not high. The authors concluded that GPx is of greatest importance in this case [32]. In another study, the authors linked changes in the activity of antioxidant enzymes in healthy men with the number of WBC treatments. The activity of individual antioxidant enzymes depended on the exposure time. Ten WBC treatments resulted in slight changes in SOD, GPx and glutathione reductase activities with a decreasing trend and a marked increase in CAT activity and glutathione levels. After 20 WBC treatments, CAT activity returned to baseline values, but SOD activity increased and GPx activity further decreased. The authors came to a very important conclusion that WBC increases oxidative stress and causes an accompanying decrease in the activity of antioxidant enzymes after 10 sessions with another compensatory increase after the end of the 20-session cycle [33]. In our study, we observed an increase of CAT in men after 20 treatments. The aim of the study by Miller et al. (2012) was to determine the effect of ten 3-min exposures on the level of total antioxidant status (TAS), the activity of selected antioxidant enzymes, uric acid (UA) and lipid peroxidation. A significant increase in plasma TAS and UA levels was observed compared to sessions without WBC. There was a statistically significant increase in SOD activity in erythrocytes obtained in the study group compared to the control group. Exposure to extremely low temperatures used in cryostimulation has been found to improve the body’s antioxidant capacity [34]. The results of the study by Mila-Kierzenkowska et al. (2013) show that WBC improves the antioxidant capacity of the body exposed to intense physical exercise (in volleyball players). The short application of cryogenic temperatures is probably related to the activation of adaptive homeostatic mechanisms according to the hormetic dose-response model [35]. Stutkowy et al. (2015) exposed subjects to extreme heat and cold. The aim of their work was to assess the impact of extreme temperatures: low (WBC) and high (dry sauna) on the oxidation-antioxidant balance in healthy men. A single treatment with low and high temperatures induced an increase in the activity of SOD and GPx (WBC) as well as SOD and CAT (dry sauna). Comparing the treatments, SOD activity was higher after a low temperature treatment was applied. Based on the research, it was concluded that both extremely high and extremely low temperatures probably cause the formation of ROS in young healthy people and thus may influence the change of the pro-oxidant-antioxidant balance [36]. The aim of the study by Stanek et al. (2016) was to assess the impact of WBC treatments on oxidative stress parameters in healthy men. The study involved 32 healthy men who were randomly divided into two groups: 16 men who underwent WBC treatments followed by kinesiotherapy and 16 men who underwent kinesiotherapy treatments only. In the WBC group, the authors observed a statistically significant decrease in the concentrations of most oxidative stress parameters with an accompanying increase in plasma concentrations of non-enzymatic antioxidants. However, they did not observe significant changes in the activity of antioxidant enzymes [31]. It has been suggested that repeated exposure to low temperatures may cause adaptive changes in the form of an increase in the total antioxidant status and the activity of selected antioxidant enzymes, which will result in the creation of a pro-oxidant-antioxidant balance at an even higher level (according to the hormesis theory) [37]. Research by Jurecka et al. (2023) shows that a single WBC treatment has no effect on the activity of antioxidant enzymes in erythrocytes in trained athletes. The researchers observed that CAT activity did not change after exercise combined with cryotherapy. Moreover, it was confirmed that the enzymatic activity of erythrocytes measured after exercise preceded by exposure to extremely low temperatures was statistically significantly lower compared to that observed after control exercise. SOD and GPx activity was higher in erythrocytes when exercise was not preceded by cryostimulation [38].

According to some authors, winter swimmers have higher baseline levels of erythrocyte-reducing enzymes (SOD, CAT and GPx) compared to controls [9,16,17]. However, swimming in cold water differs from exposure to cryogenic air temperatures, for example in that in the aquatic environment there is an additional factor of increased physical activity and hydrostatic pressure. The cold tolerance in a cryogenic chamber at very low humidity is higher than when immersed in cold water, which can be around 4 °C in the winter season. Siems et al. (1999) investigated whether repeated oxidative stress in winter swimmers results in proper antioxidant adaptation. Baseline glutathione (GSH) outputs and erythrocyte SOD and CAT activity were highest in winter swimmers. Researchers found that this improvement in antioxidant protection is due to the repetition of harmless, mild oxidative stress [9]. The aim of the research by Lubkowska et al. (2013) involved checking whether WS for five consecutive months causes adaptive changes that improve tolerance to stress caused by exposure to extremely low temperatures. During the experiment, participants were tested twice (before and after the winter season). The authors found significant changes in HGB concentration, RBC count, HCT, mean red blood cell volume and the percentage of monocytes and granulocytes after the WS season. After five months of WS, the response to cryogenic temperatures was milder. The authors concluded that the changes observed in the subjects may indicate positive adaptive changes in young healthy winter swimmers in the antioxidant system. The observed changes may increase the human body’s readiness to stress factors [39]. The effects of repeated exposure to cold and cold adaptation on human cardiovascular health are not fully understood. The aim of the study by Kralova Lesna et al. (2015) was to determine the impact of cold adaptation on cardiovascular disease risk factors, thyroid hormones and the human ability to reset the harmful effects of oxidative stress. In the group of winter swimmers, lower GPx1 activity was found compared to people from the control group. A tendency towards reduced CAT activity was also observed in the group of swimmers. Human adaptation to cold may affect markers of oxidative stress. Trends towards improved cardiovascular risk factors in cold-adapted individuals also indicate a positive impact of cold adaptation on cardioprotective mechanisms [40]. In our study, we observed changes in SOD after the winter swimming season in men and women. However, these changes were bidirectional. The difference in changes may also be influenced by different levels of estrogen. From the perspective of oxidative stress, estrogen plays a regulatory role in the cardiovascular system through the estrogen receptor, providing strategies for the treatment of menopausal women with cardiovascular diseases [41]. The aim of the research by Wesołowski et al. (2023) was to determine whether repeated exposure to cold water leads to an enhancement of antioxidant defense and whether this leads to a reduction in basal or acute impulses of oxidative dysfunction in healthy humans. Repeated exposure to low temperatures enhanced most antioxidant defense mechanisms, which led to the attenuation of basic indicators of oxidative stress and acute heart rates in response to cold exposure [42].

To identify potential sex differences in the physiological response to cold exposure, future studies could investigate hormonal, metabolic, or genetic factors that may contribute to these differences, providing valuable information on the mechanisms underlying the observed enzymatic reactions. Incorporating gene expression analyses or protein quantification techniques can provide additional insights into the mechanisms underlying the observed effects and further enhance the comprehensiveness of the study. WBC and cooling therapies are very promising methods that can reduce fatigue and improve functional status and quality of life. Cold treatments have an antioxidant effect, resulting in an improvement in the total antioxidant status in the plasma. However, there is a need for larger clinical trials with larger cohorts of participants and consistent protocols. The molecular mechanisms responsible for the effectiveness of cryotherapy and cold-water swimming in minimizing oxidative damage remain unclear. Nevertheless, hypothermia induction may play a role in partially inhibiting ROS production. The current experiment was not without flaws related to the lack of a uniform diet and its monitoring (the inclusion criterion was only the lack of change in diet before and during the project). Jiang et al. (2021) summarized the associations of dietary patterns with oxidative stress and selected metabolic diseases. High-calorie diets are one of the main factors leading to excessive production of ROS, causing inflammation, obesity and neurodegenerative diseases. On the other hand, foods rich in polyunsaturated fatty acids, polyphenols and fiber may reduce the risk of chronic diseases by regulating oxidative stress. Dietary antioxidants may protect cells against oxidative damage by neutralizing ROS [43].

## 5. Conclusions

Regular WS does not seem to place an excessive burden on the body in terms of intensifying oxidative processes. WS sessions once a week can be recommended as an effective method of improving health by causing positive adaptive changes and strengthening the body’s antioxidant barrier. WBC may be used as an adjunct therapy in the treatment of diseases caused by oxidative stress, as it improves the body’s antioxidant capacity. Further research is needed to elucidate antioxidant mechanisms in humans and to determine the short- and long-term effects of exposure to WS and WBC.

## Figures and Tables

**Table 1 biology-13-00295-t001:** General characteristics of the respondents.

Characteristics	WOMEN-WBC	MEN-WBC	(p) WOMEN-WBC/MEN-WBC	WOMEN-WS	MEN-WS	(p) WOMEN-WS/MEN-WS	WOMEN-CONT	MEN-CONT	(p) WOMEN-CONT/MEN-CONT
Age [years]	38.47 ± 5.80	35.87 ± 7.83	0.326	47.09 ± 9.15	44.30 ± 11.08	0.411	35.87 ± 7.48	30.20 ± 4.46	0.015 *
Body height [cm]	169.40 ± 5.60	179.67 ± 9.30	0.001 *	163.27 ± 5.97	179.20 ± 6.57	0.000 *	167.73 ± 7.40	182.73 ± 6.92	0.000 *
Body mass [kg]	72.35 ± 13.38	80.72 ± 11.73	0.089	65.66 ± 8.94	90.14 ± 18.15	0.001 *	66.19 ± 13.56	83.31 ± 10.90	0.000 *
Body mass index [kg/m^2^]	25.22 ± 4.65	25.06 ± 3.52	0.919	24.66 ± 3.39	27.85 ± 3.78	0.212	23.48 ± 4.27	25.01 ± 3.61	0.316
Fat [%]	30.47 ± 6.43	19.08 ± 6.55	0.000 *	30.23 ± 5.34	20.27 ± 6.32	0.000 *	25.75 ± 6.07	15.17 ± 5.60	0.000 *
Lean body mass [kg]	49.55 ± 5.70	64.96 ± 7.92	0.000 *	45.41 ± 3.55	71.16 ± 10.81	0.000 *	48.67 ± 8.00	70.23 ± 6.57	0.000 *
Total body water [kg]	36.28 ± 4.17	47.57 ± 5.80	0.000 *	33.25 ± 2.60	52.10 ± 7.93	0.000 *	35.62 ± 5.86	51.42 ± 4.81	0.000 *

* statistically significant results were considered for *p* < 0.05.

**Table 2 biology-13-00295-t002:** Intragroup comparisons of mean values of indicators—WBC.

Parameter	WOMEN-WBC			MEN-WBC		
	Study 1 (n = 15)	Study 2 (n = 15)	(p) Study 1/Study2	Study 1 (n = 15)	Study 2 (n = 15)	(p) Study1/Study2
CAT [U/g Hb]	172.32 ± 39.12	190.56 ± 50.15	0.139	122.06 ± 23.74	162.30 ± 29.23	0.000 *
GPx [U/g Hb]	49.25 ± 9.83	50.65 ± 6.42	0.586	38.05 ± 5.64	40.54 ± 4.96	0.259
SOD [U/g Hb]	1246.12 ± 147.62	1238.92 ± 126.08	0.810	1303.33 ± 152.72	1293.68 ± 173.90	0.861

* statistically significant results were considered for *p* < 0.05.

**Table 3 biology-13-00295-t003:** Intragroup comparisons of mean values of indicators—WS.

Parameter	WOMEN-WS			MEN-WS		
	Study 1 (n = 15)	Study 2 (n = 15)	(p) Study1/Study2	Study 1 (n = 15)	Study 2 (n = 15)	(p) Study1/Study2
CAT [U/g Hb]	165.16 ± 32.64	189.90 ± 58.97	0.260	158.52 ± 58.62	165.44 ± 13.32	0.446
GPx [U/g Hb]	45.29 ± 6.67	46.01 ± 6.33	0.771	39.23 ± 6.91	39.52 ± 6.29	0.906
SOD [U/g Hb]	1089.34 ± 124.94	1231.74 ± 196.06	0.004 *	1340.47 ± 275.32	1144.42 ± 122.18	0.047 *

* statistically significant results were considered for *p* < 0.05.

**Table 4 biology-13-00295-t004:** Intragroup comparisons of mean values of indicators—without intervention.

Parameter	WOMEN-CONT			MEN-CONT		
	Study 1 (n = 15)	Study 2 (n = 15)	(p) Study1/Study2	Study 1 (n = 15)	Study 2 (n = 15)	(p) Study1/Study2
CAT [U/g Hb]	189.87 ± 54.53	172.20 ± 35.71	0.156	152.86 ± 18.24	144.64 ± 23.46	0.275
GPx [U/g Hb]	50.90 ± 5.049	49.48 ± 5.63	0.431	41.19 ± 7.80	42.32 ± 6.09	0.579
SOD [U/g Hb]	1361.73 ± 202.97	1325.63 ± 173.10	0.518	1350.46 ± 195.94	1303.31 ± 118.74	0.296

**Table 5 biology-13-00295-t005:** Analysis of variance for comparisons depending on the intervention performed.

**Parameter**	**WOMEN**		**MEN**	
	F	(p)	F	(p)
CAT [U/g Hb]	1.28	0.286	6.61	0.000 *
GPx [U/g Hb]	1.47	0.219	0.66	0.619
SOD [U/g Hb]	4.28	0.003 *	2.38	0.059

* statistically significant results were considered for *p* < 0.05.

**Table 6 biology-13-00295-t006:** Post hoc test for SOD [U/g Hb] in the women’s groups.

**Group**	**WBC**	**WS**	**CONT**
**WBC**	-	0.021 *	0.062
**WS**	0.021 *	-	0.000 *
**CONT**	0.062	0.000 *	-

* statistically significant results were considered for *p* < 0.05.

**Table 7 biology-13-00295-t007:** Post hoc test for CAT [U/g Hb] in the men’s groups.

**Group**	**WBC**	**WS**	**CONT**
**WBC**	-	0.000 *	0.000 *
**WS**	0.000 *	-	0.520
**CONT**	0.000 *	0.520	-

* statistically significant results were considered for *p* < 0.05.

## Data Availability

All data generated or analyzed during this study are included in this published article.

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
