# Peer review of "The Influence of Whole-Body Cryotherapy or Winter Swimming on the Activity of Antioxidant Enzymes"

_biology, 2024, doi:10.3390/biology13050295_

Round 1

Reviewer 1 Report

Comments and Suggestions for Authors

The manuscript entitled “The influence of whole-body cryotherapy or winter swimming on the activity of antioxidant enzymes” reports that WBC may be used as an adjunct therapy in the treatment of diseases caused by oxidative stress. The topic is interesting, but the manuscript needs to be revised, and I have listed my comments below.

1. Table 1. Is each p-value obtained from the first two sets of information? For example, WOMEN-WBC and MEN-WBC. Suggest emphasizing the source of p-value in the caption.

2. Line 106-108. Suggest the author to cite some references to support the effectiveness and feasibility of the method.

3. What is meaning of "(p) 1/2"?

4. Why did the author not compare the differences and changes between WBC, WS, and CONT under the same study and gender?

5. If the author's focus is on the differences between males and females, I think the author should add some data, such as the relationship between estrogen levels and the oxidase classes detected by the author.

Author Response

Thank you for reviewing our article and your valuable suggestions to improve the manuscript.

1. signatures changed

2. references have been added to confirm the treatment methodology used

3. signatures changed

4. completely different but most commonly used WBC and CWS standard protocols are presented

5. information added

Reviewer 2 Report

Comments and Suggestions for Authors

The study conducted by the authors examines the impact of whole-body cryotherapy (WBC) and cold baths on antioxidant systems. While the research provides valuable insights, there are certain areas that require attention to enhance the quality of the study:

1.        Rationale for Antioxidant Enzymes in Blood: The authors should elaborate on why measuring antioxidant enzymes in blood serves as a reliable indicator of human health. Providing a comprehensive rationale will strengthen the significance of their findings and their implications for overall health assessment.

2.        Enhanced Descriptiveness in Results Section: The results section would benefit from more detailed descriptions to facilitate a clearer understanding of the findings. Including specific data points, statistical analyses, and comparative insights can improve the clarity and depth of the results interpretation.

3.        Gender Discrepancy in SOD Expression: The study notes a difference in Superoxide Dismutase (SOD) expression between men and women in the winter swimming intervention. An exploration into the underlying factors contributing to this gender-specific variation is necessary for a comprehensive understanding of the observed effects.

4.        Proposed Molecular Mechanisms: To elucidate the alterations in antioxidant enzyme activities following treatments, the authors are encouraged to propose potential molecular mechanisms. Integrating existing literature and scientific theories can help formulate plausible explanations for the observed changes, thereby adding depth to the study's conclusions.

5.        Expression Changes in Antioxidant Enzymes: Beyond assessing enzyme activity, it would be beneficial to examine changes in the expression of antioxidant enzymes following WBC and winter swimming interventions. Incorporating gene expression analyses or protein quantification techniques can provide additional insights into the mechanisms underlying the observed effects and further enhance the comprehensiveness of the study.

By addressing these comments, the authors can enrich the scientific discourse surrounding the effects of WBC and cold baths on antioxidant systems, ultimately contributing to advancements in understanding human health and wellness interventions.

Comments on the Quality of English Language

-

Author Response

Thank you for reviewing our article and your valuable suggestions to improve the manuscript.

1. information added

2. description has been changed, guiding the reader point by point

3. information added

4. information added

5. information added

Reviewer 3 Report

Comments and Suggestions for Authors

The manuscript “The influence of whole-body cryotherapy or winter swimming on the activity of antioxidant enzymes” investigated the impact of whole-body cryotherapy (WBC) and winter swimming (WS) on the activity of antioxidant enzymes in a group of healthy individuals. The results showed significant changes in catalase (CAT) and superoxide dismutase (SOD) levels, suggesting that these therapies may enhance the body's antioxidant defenses. These findings contribute to our understanding of the potential health benefits of WBC and WS and highlight the importance of further research in this area. Specific comments below.

1. The sentence "Oxidative stress is a neurodegenerative disorder" is misleading. Oxidative stress is not a disorder itself but a biological process linked to neurodegenerative disorders like Alzheimer's and Parkinson's. It's more accurate to say that oxidative stress is implicated in these disorders rather than being one itself.

2. While the study observed an increase in CAT activity in men after WBC, the lack of a similar response in women raises interesting questions about potential gender differences in the physiological response to cold exposure. Future research could explore hormonal, metabolic, or genetic factors that may contribute to these differences, providing valuable insights into the mechanisms underlying the observed enzyme responses. it would be valuable for the authors to consider and include this aspect in the discussion.

3. While the discussion briefly touches on the potential therapeutic applications of WBC and WS in diseases caused by oxidative stress, a more detailed discussion of the clinical implications and potential future research directions would be beneficial.

Comments on the Quality of English Language

The English language of the article is generally clear and understandable, but there are some grammar and spelling mistakes that should be proofread to improve readability. Overall, the writing of the article is appropriate and just needs minor editing and proofreading.  

Author Response

Thank you for reviewing our article and your valuable suggestions to improve the manuscript. 

1. sentence exchanged

2. information added

3. information added

Round 2

Reviewer 1 Report

Comments and Suggestions for Authors

Authors addressed the previous questions and, by doing so, improved the manuscript.

Author Response

Thank you for the information. Changes have been made in line with the Academic Editor's suggestions.